# Prevalence of Bladder and Bowel Dysfunction in Duchenne Muscular Dystrophy Using the Childhood Bladder and Bowel Dysfunction Questionnaire

**DOI:** 10.3390/life11080772

**Published:** 2021-07-30

**Authors:** Judith M. Lionarons, Imelda J. M. de Groot, Johanna M. Fock, Sylvia Klinkenberg, Desiree M. J. Vrijens, Anita C. E. Vreugdenhil, Evita G. Medici-van den Herik, Inge Cuppen, Bregje Jaeger, Erik H. Niks, Rinske Hoogerhuis, Nicky Platte-van Attekum, Frans J. M. Feron, Catharina G. Faber, Jos G. M. Hendriksen, Johan S. H. Vles

**Affiliations:** 1Department of Neurology, Maastricht University Medical Center, 6229 HX Maastricht, The Netherlands; s.klinkenberg@mumc.nl (S.K.); c.faber@mumc.nl (C.G.F.); 2School for Mental Health and Neuroscience, Maastricht University, 6229 ER Maastricht, The Netherlands; jsh.vles@mumc.nl; 3Department of Rehabilitation Medicine, Radboud University Medical Center, 6525 GA Nijmegen, The Netherlands; Imelda.deGroot@radboudumc.nl; 4Duchenne Center Netherlands, 2333 ZA Leiden, The Netherlands; e.h.niks@lumc.nl (E.H.N.); hendriksenj@kempenhaege.nl (J.G.M.H.); 5Department of Neurology, University Medical Center Groningen, 9713 GZ Groningen, The Netherlands; j.m.fock@umcg.nl; 6Department of Urology, Maastricht University Medical Center, 6229 HX Maastricht, The Netherlands; desiree.vrijens@mumc.nl; 7Department of Pediatrics, Maastricht University Medical Center, 6229 HX Maastricht, The Netherlands; a.vreugdenhil@mumc.nl; 8School of Nutrition and Translational Research in Metabolism, Maastricht University, 6229 ER Maastricht, The Netherlands; 9Department of Neurology, Erasmus University Medical Center, 3015 GD Rotterdam, The Netherlands; e.vandenherik@erasmusmc.nl; 10Department of Neurology, University Medical Center Utrecht, 3584 CX Utrecht, The Netherlands; I.Cuppen@umcutrecht.nl; 11Department of Neurology, Amsterdam University Medical Center, 1105 AZ Amsterdam, The Netherlands; b.jaeger@amsterdamumc.nl; 12Department of Neurology, Leiden University Medical Center, 2333 ZA Leiden, The Netherlands; 13Youth Healthcare Center South Limburg, 6411 TE Heerlen, The Netherlands; Rinske.Hoogerhuis@ggdzl.nl (R.H.); Nicky.Platte@ggdzl.nl (N.P.-v.A.); 14Department of Social Medicine, Maastricht University, 6229 ER Maastricht, The Netherlands; f.feron@maastrichtuniversity.nl; 15Center for Neurological Learning Disabilities, Kempenhaeghe, 5591 VE Heeze, The Netherlands

**Keywords:** Duchenne muscular dystrophy, bladder and bowel dysfunction, lower urinary tract symptoms, urinary incontinence, constipation

## Abstract

Introduction: Lower urinary tract symptoms (LUTS) and gastrointestinal (GI) problems are common in Duchenne muscular dystrophy (DMD), but not systematically assessed in regular care. We aimed to determine the prevalence of bladder and bowel dysfunction (BBD) in DMD patients compared with healthy controls (HC). Methods: The Childhood Bladder and Bowel Dysfunction Questionnaire (CBBDQ) based on the International Rome III criteria and the International Children’s Continence Society was filled out by 57 DMD patients and 56 HC. Additionally, possible associations of BBD with, for example, medication use or quality of life were evaluated in an additional questionnaire developed by experts. Results: In 74% of patients versus 56% of HC ≥ 1 LUTS (n.s.) were reported, 68% of patients versus 39% of HC reported ≥1 bowel symptom (*p* = 0.002) and 53% of patients versus 30% of HC reported combined LUTS and bowel symptoms (*p* = 0.019). A negative impact of BBD on daily life functioning was reported by 42% of patients. Conclusions: These data underscore that standard screening for BBD is needed and that the CBBDQ could be of added value to optimize DMD care.

## 1. Introduction

Duchenne muscular dystrophy (DMD) is the most common genetic neuromuscular disorder diagnosed in childhood, affecting about 1:5000 live males [1]. In DMD, X-linked recessive mutations in the *DMD* gene result in functional loss of the dystrophin protein. This dystrophinopathy is characterized by progressive striated muscle weakness, leading to loss of ambulation around the age of 12 years and eventually to fatal respiratory insufficiency and cardiomyopathy around the second or third decade [2,3]. As a result of the introduction of ventilation assistance in DMD care, survival has increased significantly during the last decades [3]. Nevertheless, these young adults with DMD often display secondary somatic symptoms, including (LUTS) lower urinary tract symptoms (i.e., urinary incontinence [4]) and gastrointestinal complaints (i.e., constipation [5]) already from a young age. These comorbidities are insufficiently investigated. Interestingly, at the cellular level dystrophin is found in smooth muscle cells and neurons of the bladder and intestine [6,7]. With progression of the disease, these comorbidities become more important and have an increasing impact on daily life functioning, therefore affecting quality of life [8].

Previous research found that 85% of DMD patients reported one or more LUTS [9]. DMD patients are in some cases even at risk for severe complications such as nephrolithiasis [10] and renal insufficiency [11,12]. However, only a small proportion of DMD patients with LUTS actually undergo urological examination and/or treatment [9]. Another common secondary symptom in DMD is constipation, which has been reported in 46.7% of patients [5] and often co-occurs with LUTS [13,14]. In addition, in some cases there has been described an increased risk for life-threatening bowel dilatation [15] and acute respiratory failure because of colonic pseudo-obstruction [16]. Moreover, bladder and bowel dysfunction (BBD) is a potential cause of significant physical and psychosocial burden for children and adults in general and their families [9,17,18,19,20,21].

Despite the high prevalence and burden of these co-occurring comorbidities, standardized screening for BBD has yet no part in the multidisciplinary standards of care guidelines for DMD [22]. In the past, others have tried to define the degree of exclusively bladder or bowel dysfunction in patients with DMD [5,9,23,24]. We determined the prevalence of both bladder and bowel dysfunction simultaneously in patients with DMD compared with healthy controls (HC).

In addition, possible associations of BBD with, for example, medication use or quality of life were evaluated. By this, we aimed to gain knowledge on the extent of BBD in DMD and to facilitate awareness for these problems, and thus screening.

## 2. Materials and Methods

### 2.1. Participants

Patients with DMD were recruited nationwide at four neuromuscular outpatient clinics in the Netherlands: Kempenhaeghe Center for Neurological Learning disabilities (CNL), Heeze, Radboud university medical center (Radboudumc), Nijmegen, University Medical Center Groningen (UMCG), Groningen and Erasmus University Medical Center (ErasmusMC), Rotterdam. Inclusion criteria were: (1) male, (2) age between 5–18 years old and (3) genetically confirmed diagnosis for DMD. Recruitment of HC occurred through the Youth Healthcare (JGZ) South Limburg at schools in Limburg, the Netherlands. Inclusion criteria for HC were: (1) male and (2) age between 5–18 years old.

Exclusion criteria for both, DMD patients and HC were: (1) intelligence quotient (IQ) <70 and (2) insufficient fluency in the Dutch language. An additional exclusion criteria for HC was a muscle disease of any kind.

### 2.2. Study Design

Patients with DMD were recruited by their treating physicians during regular visits at the neuromuscular outpatient clinic. Healthy controls were recruited during regular appointments for evaluation of general health of all school children by their youth health care professional, either at the Youth Healthcare (JGZ) clinic or at school. In the Netherlands these evaluations regularly take place at the age of five, 10, 12 and 14 years old.

The chronological time period these questionnaires were distributed was in total 12 months.

Parents/caregivers of participants between 5–12 years of age or participants of 12 years and older (with assistance from parents/caregivers if necessary) were asked to fill out questionnaires on bladder and bowel symptoms and emotional/behavioral/psychosocial functioning. Filling out the questionnaires took 15–30 min.

#### 2.2.1. Description of Participant Characteristics

Participant characteristics, i.e., age, DMD gene mutation, other diseases of any kind, neuropsychiatric/neuropsychological/neurodevelopmental disorders (i.e., attention-deficit hyperactivity disorder (ADHD), autism spectrum disorders (ASD), obsessive compulsive disorders or anxiety disorders), wheelchair dependence and medication use were asked or extracted from the electronic participant files.

The total problem score of the Dutch version of the Strengths and Difficulties Questionnaire (SDQ) for parents and youths was used as screening tool for mental health issues. This scale is based on four problem-based subscales (emotional problems, conduct problems, hyperactivity and peer problems) [25]. Total scores were standardized for age with a range of 0–40, with a cut-off score >17 for clinically significant emotional and behavioral problems [25].

#### 2.2.2. Questionnaires

##### The Childhood Bladder and Bowel Dysfunction Questionnaire (CBBDQ)

The Childhood Bladder and Bowel Dysfunction Questionnaire (CBBDQ) [26] based on the Internationally accepted Rome III criteria for functional gastrointestinal disorders [27] and the International Children’s Continence Society standardization was used to determine the prevalence of BBD [28]. The CBBDQ is an 18-item BBD symptom questionnaire, which consists of two subscales: (1) a bladder symptom scale (10 items) and (2) a bowel symptom scale (8 items) using a five-point Likert scale (0 = ‘never’ to 4 = ‘almost daily or daily’). To examine the symptom prevalence rates of the CBBDQ, the outcomes were dichotomized with ‘never or once in the preceding month (0–1)’ scored as non-symptomatic and ‘more than once to (almost) every day in the preceding month (2–4)’ as symptomatic (Appendix A).

##### Supplementary BBD Questionnaire Developed by Physicians

A supplementary 33-item BBD questionnaire developed by a urologist (D.V.) and pediatric gastroenterologist (A.V.) was used to assess symptoms that were not asked in the CBBDQ (Appendix A). This questionnaire contained additional items on urinary continence (2 items) and fecal continence (1 item), history of bladder (8 items) and bowel dysfunction (8 items), micturition (6 items) and defecation pattern (3 items), toilet use and aids (3 items), relationship between bladder- and bowel dysfunction (1 item), and influence on daily life functioning (1 item). Influence on daily life functioning of these problems was rated on a scale from 0 to 10 (‘no bother at all’ to ‘very bothersome’).

For all questions a description was given of the prevalence rates. A proportion of the questions was rated on a four-point Likert scale: ‘never’ (0), ‘sometimes’ (1), ‘often’ (2), ‘always’ (3) in the preceding month.

### 2.3. Statistical Analysis

Differences in age and SDQ total problem scores between DMD patients and HC were assessed using independent samples t-tests. Differences between DMD patients and HC in symptomatic scores (2–4) on the CBBDQ were assessed using Pearson’s chi-square tests. A possible correlation between the prevalence of at least one bladder symptom and at least one bowel symptom was assessed using a linear regression model for both DMD patients and HC. Additionally, potential effects of age in DMD patients and HC, and wheelchair dependence and steroid use in DMD patients on symptomatic scores on the CBBDQ were assessed using a linear regression model. Possible multiple comparisons were corrected for using the method of Benjamini and Hochberg, which limits the false discovery rate to 5% [29]. All group comparisons were considered significant at *p* < 0.05. All analyses were performed in SPSS Statistics Version 25 (IBM, Armonk, NY, USA).

## 3. Results

### 3.1. Participant Characteristics

Fifty-seven boys with DMD (age range = 5–18 years, *n*= 23 by proxy and *n* = 34 self-report) and 56 HC (age range = 5–18 years, *n* = 30 by proxy and *n* = 26 self-report) participated in this study. None of the participants and/or their parents/caregivers had to be excluded due to insufficient answering even though they were initially eligible to participate in this study. An overview of the participant characteristics is given in Table 1. We found a higher median age in DMD compared with HC (*p* = 0.006). Total problems scores on the SDQ were within the normal range (<17) for both DMD (*N* = 69; *n* = 38 by proxy and *n* = 31 self-report) and HC (*N* = 73; *n* = 46 by proxy and *n* = 27 self-report) and no differences were found between the DMD and HC group or between by proxy and self-report.

### 3.2. CBBDQ

In DMD, 51 of the 57 DMD (89%) patients experienced at least one bladder and/or bowel symptom (symptomatic score on the CBBDQ; at least one bladder symptom (*n* = 42) + at least one bowel symptom (*n* = 39)—combined bladder and bowel symptom (*n* = 30)).

A summary of the prevalence of parent- and self-reported bladder and bowel symptoms based on the CBBDQ is presented in Table 2. Several LUTS indicating storage problems were reported more often in DMD compared with HC: (intermittent) daytime urinary incontinence (item 2 and 4; *p* = 0.001 and *p* = 0.026 respectively), urgency (item 7; *p* = 0.037) and nocturia (item 10; *p* = 0.020). The prevalence of at least one bladder symptom was comparable for DMD and HC. The median age for a symptomatic bladder symptom score (score of 2–4 on items 1–10) was, except for dribbling (item 3), higher in DMD compared with HC. As we found a higher median age in the total DMD group compared with HC (*p* = 0.006), it is not possible to differentiate if the higher median age of a symptomatic bladder symptom score is a result of the overall higher age of the cohort or if bladder symptoms are more common at a higher age for DMD patients.

Two or fewer bowel movements per week (constipation) and abdominal pain were more prevalent in DMD compared with HC (item 11 and 17; *p* = 0.010 and *p* = 0.032 respectively). The prevalence of at least one bowel symptom was higher in DMD compared with HC (*p* = 0.002). The median age for a symptomatic bowel symptom score (score of 2–4 on items 11–18) was, except for hard stools or painful bowel movements (item 13), higher in DMD compared with HC. Additionally, for a symptomatic bowel symptom score, it is not possible to differentiate if the higher median age of symptoms is a result of the overall higher age of the cohort or if bowel symptoms are more common at a higher age for DMD patients.

Two or fewer bowel movements per week (item 11) were reported with a particularly high median age of 16.0 in DMD. The occurrence of bladder and bowel symptoms simultaneously was reported more often in DMD compared with HC (combined bladder and bowel symptom; *p* = 0.019). When looking into a possible correlation between the prevalence of at least one bladder symptom and at least one bowel symptom, we found a positive correlation in HC (*p* = 0.025), but not in DMD.

In DMD, additional regression analyses of CBBDQ items in relation to age showed an age effect for item 9 ‘wets the bed or diaper during sleeping periods’ (*p* = 0.046); the lower the age this item was reported more frequently. In HC, an age effect was found for the items 8 ‘postpones first urination in the morning’ and 16 ‘experiences of a sudden uncontrollable urge to defecate’ (*p* = 0.043 and *p* = 0.030 respectively); the lower the age these items were reported more frequently. For the remaining items, no age effects were found for both DMD and HC. Furthermore, regression analyses of CBBDQ items in relation to wheelchair dependence showed that the prevalence of at least one bladder symptom in general (not one specific bladder symptom) was higher in wheelchair dependent DMD patients (*p* = 0.032). For the remaining items, no association of wheelchair dependence was found in DMD. Finally, regression analyses of CBBDQ items in relation to steroid treatment showed that the prevalence of nocturia was higher in patients who were treated with steroids (item 10; *p* = 0.048). Two or fewer bowel movements per week were also more frequently reported in relation to steroid treatment (item 11; *p* = 0.015). For the remaining items, no relation with steroid treatment was found in DMD.

### 3.3. Supplementary BBD Questionnaire Developed by Physicians

Prevalence of parent- and self-reported bladder and bowel symptoms based on the supplementary BBD questionnaire developed by physicians, which were scored on a Likert scale are presented in Table 3. The remaining questions are described in the Section 3.3.1 and Section 3.3.2.

#### 3.3.1. Bladder Symptoms

Except for dysuria, LUTS indicating voiding problems such as hesitancy, straining, weak stream and intermittency appeared to be more prevalent in DMD compared with HC (Table 3). The median age for bladder symptoms was, except for urgency, higher in DMD compared with HC (Table 3). In DMD, four patients reported daytime and night time urinary incontinence simultaneously (*n* = 1 five-year-old, *n* = 1 six-year-old, *n* = 1 seven-year-old, *n* = 1 11-year-old) and two patients reported only night time urinary incontinence (*n* = 1 eight-year-old and *n* = 1 10-year-old). In HC, all participants were continent for urine during the day at the time of this study. Three HC reported night time urinary incontinence (*n* = 2 five-year-old and *n* = 1 eight-year-old). Daytime urinary continence and night time urinary continence had been achieved between the age of two to four years old for the majority of participants (77% DMD and 88% HC). In three patients, a urologist was involved on the indication of storage problems (*n* = 1) and voiding problems (*n* = 2) and in one patient a pediatrician on the indication of storage problems. In two of these patients, the treating urologist performed extensive urological evaluation, including a bladder diary, physical examination, uroflowmetry and a bladder ultrasound. Reported treatment of bladder symptoms included toilet training (*n* = 3) and anticholinergics (oxybutynin) on the indication of an overactive bladder (*n* = 1), resulting in symptom improvement. In one HC, a general practitioner was involved because of recurrent urinary tract infections.

#### 3.3.2. Bowel Symptoms

Except for anal fissures, bowel symptoms (blood in stool and loss of sensation of passing stools) appeared to be more prevalent in DMD compared with HC (Table 3). The median age for bowel symptoms was higher in DMD compared with HC (Table 3).

In DMD, the same four patients, who reported daytime urinary incontinence also reported fecal incontinence. Fecal continence had been achieved between the age of two to four years old for the majority of participants (67% DMD and 79% HC). In four patients, a pediatrician was involved in the indication of constipation (*n* = 2), fecal incontinence (*n* = 1) and an irregular defecation pattern alternating between both constipation and fecal incontinence (*n* = 1). In the one patient with an irregular defecation pattern gastrointestinal examination of the contraction of the external anal sphincter was performed by the treating pediatrician. Reported treatment of bowel symptoms included a fiber rich diet (*n* = 6), toilet training (*n* = 2), physiotherapy (*n* = 1), daily probiotics (*n* = 1) and laxatives on the indication of constipation (*n* = 15), resulting in symptom improvement. In the two HC, who never attained fecal continence a general practitioner (*n* = 1) and pediatrician and psychologist (*n* = 1) were involved. General physical examination of the abdomen took place in both of these HC. Reported treatment included toilet training (*n* = 2) and one of these patients also maintained a fiber rich diet.

#### 3.3.3. Aids Required

Sixteen DMD patients used the toilet independently (median age = 14.0 years, age range = 5–18 years). The other 41 DMD patients needed assistance. For 34 of the DMD patients, toilet aids were required to use the toilet (median age = 14.0 years, age range 5–18 years). The most frequently reported aids were a hoist, toilet brackets, toilet seat adjustments (i.e., toilet seat elevation/reduction/elongation), a commode chair, a chamber pot and a toilet shower with or without an air dryer. All 56 HC used the toilet independently without the requirement of any toilet aids.

#### 3.3.4. Relationship between Bladder and Bowel Symptoms and Quality of Life

Forty-two percent of patients reported a negative influence of bladder and/or bowel symptoms on daily life functioning. Patients who did report a negative influence on daily life functioning rated these problems with a mean score of 3.0 (range 1–8) on scale from 0–10. The median age of patients, who reported a negative influence on daily life functioning was 14.0 years (age range = 6–18). Reported consequences of bladder and/or bowel symptoms causing these negative effects on daily life were: dirty smell because of stool stains/soils in underwear, discomfort or stomach ache because of constipation, use of a diaper, difficulty finding a suitable changing spot outside of the house, few people are willing to change a diaper of a grown child in private, often having to leave the classroom to urinate, accidently wetting themselves during the day without wearing a diaper.

Thirteen percent of HC reported a negative influence of bladder and/or bowel symptoms on daily life functioning. The HC who did report a negative influence on daily life functioning rated these problems with a mean score of 2.4 (range 1–6) on scale from 0–10. The median age of HC, who reported a negative influence on daily life functioning was 7.5 years (age range = 5–15). Reported consequences of bladder and/or bowel symptoms causing these negative effects on daily life were: difficulty holding stools during school, stool stains in underwear, decreased self-confidence, often having to leave the classroom to urinate and waking up with a wet pants in the morning.

For quality of life, there were no differences found between by proxy and self-report.

## 4. Discussion

We aimed to gain knowledge on the extent of BBD in DMD by determining the prevalence of BBD in patients with DMD compared with HC using the CBBDQ. In addition, we assessed the impact of BBD on daily life functioning and evaluated possible associations of BBD with potential contributing factors, such as wheelchair dependence and steroid use using a supplementary questionnaire developed by experts. The main findings of this study are that: (1) BBD was more prevalent in DMD patients compared with HC; (2) Bladder dysfunction coincides more often with bowel dysfunction in DMD patients compared with HC; (3) wheelchair dependence was associated with bladder dysfunction, but not with bowel dysfunction; (4) steroid treatment was associated with nocturia and constipation; and (5) a negative impact of BBD on daily life functioning was reported by 42% of patients.

### 4.1. Prevalence of BBD in DMD

In this study, the prevalence of LUTS found using the CBBDQ was in between previously reported prevalence rates from 50% up to 85% of DMD patients [9,17,23,32]. Based on these results, the CBBDQ has been introduced as a simple and convenient way of systematically evaluating BBD in children in *N* = 1748 children from 1–16 years of age through parent report, irrespective of the cause or presence of comorbidities and/or behavioral problems [33].

Notably, we did not observe gross changes in the prevalence of certain bladder symptoms between younger and older DMD patients. Van Wijk and colleagues previously found that younger DMD patients experienced mostly storage LUTS, while older DMD patients had more problems during voiding and post micturition [9]. Our data did not confirm these findings, as the median age for storage and voiding problems in DMD patients was comparable in this study. The reason we did not detect clear age differences in bladder symptoms might be due to the smaller age range and sample size of our DMD cohort.

In addition to the current literature, we compared the prevalence of LUTS in DMD patients with HC. Several LUTS indicating storage problems were reported more often in DMD compared with HC using the CBBDQ: (intermittent) daytime urinary incontinence, urgency and nocturia. Although the prevalence of at least one bladder symptom was comparable for DMD and HC, the median age for a symptomatic bladder symptom score was overall higher in DMD compared with HC. This might suggest that HC outgrow storage problems, which are frequently reported at a young age in the general population, whereas these problems persist in DMD patients as they grow older. It is well-known that night time urinary incontinence can be temporary due to young age [34]. With additional regression analyses we found minimal age effects. In HC, postponing the first urination in the morning was more often reported at a younger age and in DMD, enuresis was reported more often at a younger age. For the remaining items, no age effects were found for both groups. As we found a higher median age in the total DMD group compared with HC (*p* = 0.006), it is not possible to differentiate if the higher median age of a symptomatic bladder symptom score is a result of the overall higher age of the cohort or if bladder symptoms are more common at a higher age for DMD patients.

On the other hand, the prevalence of constipation in DMD patients found in this study was lower than previously reported (46.7%) [5]. This might suggest that complaints of constipation were successfully treated in a greater proportion of our DMD cohort with fiber rich diet, toilet training, physiotherapy, probiotics and laxatives. Compared with HC, we found that DMD patients in this study more often reported bowel dysfunction in general and two or fewer bowel movements per week (constipation) and abdominal pain using the CBBDQ. Overall, the median age for a symptomatic bowel symptom score was higher in DMD compared with HC. However, we solely found an age effect for the experience of a sudden uncontrollable urge to defecate in HC. In DMD, no age effects were found for bowel symptoms. These findings confirmed previous research, which also did not find a significant increase in the prevalence of constipation with age or with worse functional status [5]. Additionally, for a symptomatic bowel symptom score, it is not possible to differentiate if the higher mean age of symptoms is a result of the overall higher age of the cohort or if bowel symptoms are more common at a higher age for DMD patients.

With regards to functional status, we looked into the relationship between BBD and wheelchair dependence. This sub-analysis showed that wheelchair dependence correlated with bladder dysfunction in general, but not with bowel dysfunction. For the majority of wheelchair dependent DMD patients, personal assistance is required to use the toilet. This means they frequently need to ask for help, which may be a threshold for some patients. When transitioning to adulthood with a progressive disease, adolescent DMD patients report the desire to be as independent as possible and have previously reported that dependence on parents would be less acceptable as they grow older [35].

The final sub-analysis showed that steroid treatment was associated with nocturia and constipation. This is in line with previous literature, as long-term steroids use is known to have an important impact on nocturia [36] due to reversal of the normal circadian rhythm of water and electrolyte excretion resulting in a decrease in proximal tubular resorption of sodium at night [37]. Additionally, long-term steroid use can promote pathology in the large intestine due to toxic injury or vascular insufficiency [38,39].

### 4.2. The Relationship between BBD and Possible Explanations for BBD in DMD

We found that bladder and bowel symptoms were more often reported simultaneously in DMD patients compared with HC. Nonetheless, we did not find a relation between bladder and bowel dysfunction in DMD patients, while we did in HC. A review on the relationship between constipation and LUTS, found that several studies in children documented an association without a clear explanation on underlying pathophysiology [40]. The available data on patients with neuropathic lower urinary tract and colorectal dysfunction, however, suggested that stool impaction in the rectum may mechanically hamper bladder emptying [41]. This could for a part be a possible explanation for the coincidence of bladder and bowel dysfunction in DMD. Except when symptoms of BBD clearly result from severe scoliosis or spinal surgery [42,43], the neurological basis of BBD in DMD remains ambiguous. Other central- (i.e., potential dysfunctional expression of dystrophin in the spinal cord [44]) as well as peripheral disturbances (i.e., skeletal muscle weakness of the pelvic wall, smooth muscle weakness in the urinary and gastrointestinal tract [32] or potential dysfunctional afferent nerve signaling in the bladder [6]) could play a role in the occurrence of BBD. Additionally, a combination of these factors and/or behavioral problems may contribute to BBD. The voiding problems, which were reported in this study, such as hesitancy, straining, weak stream and intermittency could be caused by potential weakness of the bladder wall in DMD. This can result in an incomplete emptying and possibly overflow incontinence, which has previously been reported in patients with a neurogenic bladder [45]. A similar mechanism might be present in DMD. The presence of an overactive bladder in DMD is more difficult to explain from pathophysiological point of view. Future research should include urodynamic examination to further look into this [46]. Nocturia, or nocturnal polyuria in specific, could have multiple causes. Nutrition, and in particular (in)sufficient fiber and water consumption, as well as commonly used medication in DMD (i.e., steroids, diuretics and angiotensin-converting-enzyme inhibitors; [22,47]) play a role in the occurrence of nocturia in general. Additionally, bladder problems such as a small bladder capacity or an overactive bladder could lead to nocturia [36]. In DMD, both cardiomyopathy [48], as well as sleep disorders [49] could be a possible cause of nocturia [36]. Furthermore, it is well-known that immobilization in general can result in constipation [50]. Little is known about the role of immobilization in bladder dysfunction. In children with spinal muscular atrophy, however, it has been suggested that voiding postponement and dysfunctional voiding may be habitual due to their motor incapacitation [51].

Although, general mental health was overall good in this study. As total problems scores on the SDQ were within the normal range (<17) for both DMD and HC and no differences were found between the DMD and HC group or between by proxy and self-report. Life events in DMD as a consequence of the disease and its progression, such as loss of ambulation, frequent hospital visits may also play a role in the occurrence of BBD. A large number of life events (e.g., illness, injury, death, family composition or living conditions), to which children in general have difficulty adapting can be of importance for the age of acquisition of bladder control. Jansson and colleagues showed that the more life events and the older the child was when experiencing them, the later the child became dry [52]. Behavioral aspects, such as delaying urination by not wanting to ask for help or whilst being distracted playing or watching television in boys and adolescents can result in holding maneuvers, low voiding frequency, urgency and daytime incontinence. Constipation is often associated with voiding postponement [53].

### 4.3. Impact of BBD on Quality of Life

In this study, a negative influence of bladder and/or bowel symptoms on daily life functioning was reported in 42% of DMD patients, which is comparable with previous literature on the psychosocial impact of LUTS in DMD [9]. There were no differences found between by proxy and self-report. Notably, the median age of participants reporting a negative impact of BBD on daily life functioning appeared to be higher in DMD patients (14.0) compared with HC (7.5), which could underscore the importance of good urological and gastroenterological follow-up and treatment when necessary in multidisciplinary DMD care. However, as we found a higher median age in the total DMD group compared with HC, it is not possible to differentiate if a higher median age of a negative impact of BBD on daily life is a result of the overall higher age of the cohort or if a negative impact of BBD on daily life is more common at a higher age for DMD patients.

### 4.4. Limitations

The most impactful limitations of this study were the small sample size, and the higher median age in the total DMD group compared with HC. Therefore it was not possible to differentiate if the higher median age in the prevalence of symptoms is a result of the overall higher age of the cohort or if symptoms are more common at a higher age for DMD patients. Furthermore, the CBBDQ is developed to evaluate BBD in children from 5–12 years of age based on International guidelines for children [25]. This makes this screening tool less suitable for older DMD patients, as important voiding and post-micturition problems which are frequently seen in older DMD patients are not represented in this questionnaire. Additional questions on these problems, should be asked by the treating physician. Additionally, a large number of questionnaires have been filled in by parents or caregivers. As previous research found that parents may interpret questions involving quality of life differently than patients, our findings might not fully reflect patients’ experiences on BBD [54]. Lim and colleagues found that 1 out of 8 items on the physical health scale and 3 out of 15 items on the psychosocial health scale of the pediatric quality of life inventory versions 4.0 significantly differed between patients with DMD and their parents [54]. In addition, parents of children with chronic disease may be more attuned to their child’s daily health habits than parents of HC, which may have affected our results. However, there were no differences found between by proxy and self-report of quality of life for our cohort. Finally, the factors age and wheelchair dependence, and wheelchair dependence and steroid treatment were related to each other and are no independent factors, which potentially affected our regression analyses.

### 4.5. Clinical Implications/Recommendations

Based on our findings, bowel dysfunction appears to already be recognized and successfully treated in some DMD patients. On the other hand, bladder dysfunction in general was often unrecognized and untreated. Thus, there remains room for improvement in the diagnosis and treatment of BBD in DMD, especially in bladder dysfunction. We recommend standard screening for BBD in all DMD patients. The CBBDQ may have added diagnostic value to facilitate screening. The CBBDQ has been validated in a large population of children (*N* = 1333, aged 5–12 years old) visiting (pelvic) physiotherapy, affected with at least one bladder or bowel dysfunction; Cronbach alpha values were 0.74 and 0.71 for bladder and bowel subscales, respectively [26].

In the case of LUTS, an extensive urological evaluation (including urodynamic examination) and a thorough follow-up is recommended. Standard screening for BBD from an early age could facilitate the diagnosis of possible LUTS or gastrointestinal problems at a later stage of the disease, and will eventually optimize multidisciplinary DMD care. A longitudinal study design is recommended to assess whether BBD management is associated with improved quality of life. This is an important issue in gaining or maintaining autonomy for DMD patients, as they grow older.

## 5. Conclusions

BBD is more prevalent in DMD patients compared with HC. Constipation was treated in a proportion of DMD patients, but LUTS often remained unrecognized and untreated. We recommend standard screening for BBD as part of multidisciplinary DMD care for all patients. The CBBDQ may add diagnostic value to facilitate screening. As important voiding and post-micturition problems which are frequently seen in older DMD patients are not represented in the CBBDQ, additional questions on this should be asked by the treating physician. To this end, the questions that were asked in the supplementary questionnaire in this study could be used. Regular follow-up of possible BBD is needed, as patients grow older. Future research should also include urodynamic examination, to gain a better understanding of the underlying pathophysiology of BBD in DMD to facilitate more specific treatment.

## Figures and Tables

**Table 1 life-11-00772-t001:** Participant characteristics for all sites. ACE = angiotensin-converting-enzyme; ADHD = attention-deficit hyperactivity disorder; ASD = autism Spectrum Disorders; CNL= Kempenhaeghe Centre for Neurological Learning disabilities; DMD = Duchenne muscular dystrophy; ErasmusMC = Erasmus University Medical Centre; NAO = Not otherwise specified; Radboudumc = Radboud university medical center; SDQ = strength and difficulties questionnaire for emotional and behavioral problems; UMCG = University Medical Centre Groningen. The following DMD mutations were classified: (1) mutations upstream of intron 44 were considered Dp140+; (2) mutations involving the transcription start site in intron 44 or involving the genomic region downstream of exon 51 were considered Dp140—[30]; (3) mutations downstream of exon 63 were considered Dp71—[31]; and (4) mutations in-between exon 45 and 50, from which expression of Dp140 could not be predicted.

Characteristics	DMD	HC
	Total	Total
Participants, *n*	57	56
Age, yr, median (range)	13.0 (5–18)	10.0 (5–18)
DMD mutations, *n*		
Exon 1–44 (Dp140+)	27	*-*
Exon 51–62 (Dp140−)	16	*-*
Exon 63–79 (Dp71−)	2	*-*
Intron 44–exon 50 (Undefined)	7	*-*
Urogenital/gastrointestinal problems, *n*		
Small bladder capacity	1	-
Urinary tract infection	-	1
Phimosis	2	-
Cryptorchidism	-	2
Hypospadias	1	1
Kidney stones	1	-
Hydronephrosis	-	1
Surgery, *n*		
Scoliosis	10	-
Bladder reduction	1	-
Circumcision	3	1
Cryptorchidism repair	-	2
Hypospadias repair	1	1
Inguinal hernia	3	-
Fractures (e.g., hip)	2	-
Medication, *n*		
Laxatives	15	-
Anticholinergics		
Oxybutynin	1	-
Diuretics	2	-
ACE inhibitors	14	-
Beta2-sympaticomimetics/inhaled	1	5
corticosteroids		
Psychopharmaceuticals		
Methylphenidate	3	1
Fluoxetine	1	-
Aripiprazole	1	-
Steroid treatment, *n*	42	-
Wheelchair bound, *n*	47	-
Neuropsychiatric, neurodevelopmental or learning disorders, *n*		
ADHD	4	1
ASD	4	1
Anxiety disorder	1	1
Dyslexia	1	3
Learning disorder NAO	*-*	1
SDQ total problems score, mean (SD)	8.9 (5.3)	6.6 (5.0)
Parent report	9.4 (5.1)	6.5 (5.5)
Self-report	8.3 (5.5)	6.6 (4.5)

**Table 2 life-11-00772-t002:** Differences in childhood bladder and bowel dysfunctions (parent- and self-reported) between DMD and control group. Data are presented as *n* (%) with median age (range). ^a^ Likert scale for symptoms on all items are (never-(nearly) every day) dichotomized: ‘Never or once in the preceding month’ classified as ‘non-symptomatic’ (no); ‘more than once in the preceding month to (nearly) every day in the past month’ classified as ‘symptomatic’ (yes). Missing items were imputed as ‘non-symptomatic’. ^b^ Pearson’s chi-square; * *p*-value < 0.05. DMD = Duchenne muscular dystrophy; HC = healthy control.

Items ^a^	DMD		HC		*p*-Value ^b^
	*N* = 57	Median age (range), yr	*N* = 56	Median age (range), yr	
1. Passes urine more than 8 times during the day	8 (14)	9.0 (6–16)	2 (4)	7.5 (5–10)	0.062
2. Wets underwear and/or outer clothing during the day	20 (35)	13.0 (6–18)	5 (9)	10.0 (5–18)	0.001 *
3. Loses some drops of urine immediately after urinating has finished	18 (32)	14.0 (6–18)	11 (20)	14.0 (5–18)	0.161
4. Loses urine within the hour after urinating has finished	5 (9)	10.5 (7–14)	0 (0)	-	0.026 *
5. Seems to ignore the urge to urinate	16 (28)	13.0 (6–18)	11 (20)	7.0 (5–15)	0.293
6. Uses tricks to stay dry, like wriggling or forcefully crossing the legs	15 (26)	9.0 (6–17)	9 (16)	8.0 (5–16)	0.182
7. Experiences a sudden uncontrollable urge to urinate	12 (21)	13.0 (6–17)	4 (7)	8.0 (6–13)	0.037 *
8. Postpones first urination in the morning	4 (7)	10.0 (6–14)	4 (7)	6.0 (5–10)	0.936
9. Wets the bed or diaper during sleeping periods	6 (11)	8.0 (6–16)	3 (5)	8.0 (5–10)	0.310
10. Wakes up at night to urinate	16 (28)	13.0 (8–18)	6 (11)	10.0 (5–12)	0.020 *
11. Has two or fewer bowel movements per week	9 (16)	16.0 (5–18)	1 (2)	13.0 (13)	0.010 *
12. Stains or soils the underwear with stools	13 (23)	12.0 (6–18)	7 (13)	9.0 (5–13)	0.150
13. Has hard stools or painful bowel movements	6 (11)	11.5 (7–17)	4 (7)	13.5 (7–14)	0.507
14. Has large amounts of stool (that may obstruct the toilet)	5 (9)	8.5 (8–18)	3 (5)	6.0 (5–14)	0.513
15. Postpones bowel movements	12 (21)	12.5 (6–17)	8 (14)	7.0 (5–18)	0.369
16. Experiences a sudden uncontrollable urge to defecate	8 (14)	9.5 (6–17)	3 (5)	6.0 (5–7)	0.112
17. Has abdominal pain	16 (28)	13.5 (8–18)	6 (11)	8.5 (7–16)	0.032 *
18. Has a bloated belly	8 (14)	14.5 (8–17)	4 (7)	8.5 (5–13)	0.247
At least one bladder symptom	42 (74)	13.0 (6–18)	32 (57)	10.0 (5–18)	0.079
At least one bowel symptom	39 (68)	14.0 (5–18)	22 (39)	12.5 (5–18)	0.002 *
Combined bladder and bowel symptom	30 (53)	13.0 (6–18)	17 (30)	12.0 (5–18)	0.019 *

**Table 3 life-11-00772-t003:** Bladder and bowel dysfunctions (physicians’ questions). Data are presented as *n* (%) with median age (range). ^a^ Likert scale for symptoms on all items are ‘never’ (0), ‘sometimes’ (1), ‘often’ (2), ‘always’ (3) in the preceding month. Missing items were imputed as ‘never’. DMD = Duchenne muscular dystrophy; HC = healthy control.

Items ^a^	DMD		HC	
	Total	Median age (range), year	Total	Median age (range), year
	*N* = 57		*N* = 56	
**Hesitancy**				
Sometimes	25 (44)	13.0 (6–18)	16 (29)	12.5 (5–18)
Often	3 (5)	15.0 (14–16)	0 (0)	-
Always	0 (0)	-	0 (0)	-
**Straining**				
Sometimes	18 (31)	15.5 (7–18)	6 (11)	12.5 (6–15)
Often	1 (2)	14.0 (14)	0 (0)	-
Always	0 (0)	-	0 (0)	-
**Intermittency**				
Sometimes	9 (16)	13.0 (7–18)	2 (3)	12.0 (12)
Often	0 (0)	-	0 (0)	-
Always	0 (0)	-	0 (0)	-
**Dysuria**				
Sometimes	6 (10)	13.0 (8–15)	7 (12)	8.0 (5–16)
Often	0 (0)	-	0 (0)	-
Always	0 (0)	-	0 (0)	-
**Urgency**				
Sometimes	15 (26)	13.0 (8–18)	10 (18)	11.5 (6–14)
Often	1 (2)	8.0 (8)	3 (5)	10.0 (10–13)
Always	0 (0)	-	0 (0)	-
**Weak stream**				
Sometimes	8 (14)	14.0 (7–18)	5 (9)	13.0 (5–15)
Often	2 (3)	12.0 (10–14)	0 (0)	-
Always	1 (2)	14.0 (14)	0 (0)	-
**Blood in stools**				
Sometimes	4 (7)	14.0 (7–18)	1 (2)	8.0 (8)
Often	1 (2)	17.0 (17)	0 (0)	-
Always	0 (0)	-	0 (0)	-
**Anal fissures**				
Sometimes	2 (4)	11.0 (7–15)	3 (5)	6.0 (5–8)
Often	0 (0)	-	0 (0)	-
Always	0 (0)	-	0 (0)	-
**Reduced sensation of passing stools**				
Sometimes	5 (9)	13.0 (5–18)	0 (0)	-
Often	0 (0)	-	0 (0)	-
Always	2 (4)	7.5 (7–8)	0 (0)	-

## Data Availability

The data presented in this study are available on request from the corresponding author. The data are not publicly available by choice of the research team.

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
