# Peer review of "Prevalence of Bladder and Bowel Dysfunction in Duchenne Muscular Dystrophy Using the Childhood Bladder and Bowel Dysfunction Questionnaire"

_life, 2021, doi:10.3390/life11080772_

Round 1

Reviewer 1 Report

This is an interesting paper on a very clinically-relevant topic for which there is little data.  The novel questionnaire may be a very good screening tool for physicians. Seeing these data, neurologists may be prompted to query their patients about these symptoms, as they may not often be volunteered. 

  • Most weaknesses of the study are appropriately stated in the paper, the most impactful being the small sample size, and the  higher median age in the total DMD group compared with HC (p = .006). Therefore it is not possible to differentiate if the higher median age of any symptom is a result of the overall higher age of the cohort or if symptoms are more common at a higher age for DMD patients.
  • Seeing the lack of association between bowel and bladder symptoms is not surprising because (as described in the paper) this is usually the result of stool withholding. Stool withholding typically develops at younger ages than studied here. In addition, there was a lack of other symptoms suggestive of withholding (encopresis, large volume of stool, soiling).  However, I would then remove this sentence (page 13, line 423), as it is not supported by the results: In addition, life events can also be the cause of voluntary retention of faeces [54]. 
  • Some minor wording change: 
    • Page 2, line 51, Change "Even though" isn't grammatically correct in English. Consider "Nonetheless' or "Nevertheless"
    • Page 9, line 275: comma should be moved to follow "continence" in the next line.
  • As it is acknowledged that many items were parent-reported and that patient responses may be different, it should also be acknowledged that parents of children with a chronic disease may be more attuned to their child's daily health habits than parents of control participants. 

Author Response

Response to Reviewer 1 Comments

Point 1: Most weaknesses of the study are appropriately stated in the paper, the most impactful being the small sample size, and the  higher median age in the total DMD group compared with HC (p = .006). Therefore it is not possible to differentiate if the higher median age of any symptom is a result of the overall higher age of the cohort or if symptoms are more common at a higher age for DMD patients.

Response 1: We acknowledge this comment, and already stated these weaknesses in the ´Prevalence of BBD in DMD´ section. For the revision, this is emphasized by adding two statements about this to the limitations section of the manuscript (p. 17, first to lines of this section).

Point 2: Seeing the lack of association between bowel and bladder symptoms is not surprising because (as described in the paper) this is usually the result of stool withholding. Stool withholding typically develops at younger ages than studied here. In addition, there was a lack of other symptoms suggestive of withholding (encopresis, large volume of stool, soiling).  However, I would then remove this sentence (page 13, line 423), as it is not supported by the results: In addition, life events can also be the cause of voluntary retention of faeces [54]. 

Response 2: We acknowledge this comment, are thankful for the suggestion and agree that this sentence should be removed, as our study results do not confirm these previous findings. 

Point 3: Some minor wording change: 

  • Page 2, line 51, Change "Even though" isn't grammatically correct in English. Consider "Nonetheless' or "Nevertheless"
  • Page 9, line 275: comma should be moved to follow "continence" in the next line.

Response 3: We acknowledge these comments, and changed the wording as suggested on page 2: ´Even though´ is changed into ´Nevertheless´ on page 2, line 51. About the comma that is suggested to be moved on page 9, I am not sure what the reviewer means. Could you maybe further explain, so I can adjust this?

Point 4: As it is acknowledged that many items were parent-reported and that patient responses may be different, it should also be acknowledged that parents of children with a chronic disease may be more attuned to their child's daily health habits than parents of control participants.

Response 4: we are thankful for this comment and agree that this is important to mention in the manuscript. Therefore, we added a statement about this to the last part of the Limitations section.

Reviewer 2 Report

This manuscript describing the prevalence of bladder and bowel dysfunction in children with Duchenne Muscular Dystrophy, comparing it to age-matched healthy controls is well written. It describes many of the facets of the dysfunction in these two organ systems following questionnaires given to children and their families. There are a few questions I have that when answered should improve the clarity and scientific quality of their communication.

  1. The abstract should be ‘structured’ with an Introduction, Methods, Results and Conclusions sections.
  2. Much of the 3rd paragraph in the Introduction of the main text (lines 72 – 86) can be shortened and transferred to the Methods section with the last part placed in the Discussion section.
  3. In the Methods section under study design (2.2) please describe what chronological time period these questionnaires were distributed, exactly how the children with DMD and healthy controls were recruited initially, how many declined to participate, how many patients and families had to be excluded due to insufficient answering of the questions even though they were initially eligible?
  4. How many patients answered the questionnaires themselves and how many caregivers had to respond to the questionnaires?
  5. In section 3.3.3 of the Results, I am not sure which group (DMD or HC or both) the first two sentences refer to. In sentence 2, “The other 31…” refers to which group? Can the authors clarify this point?
  6. Additionally, the number of patients quoted by the authors in this paragraph do not add up to the 57 DMD patients or the 56 healthy controls. Can the authors provide clarity to this point?
  7. In section 3.3.4 of the Results, were there any age adjustment differences noted between the two groups and if so, please describe them in a revised version.
  8. In the Discussion section, under prevalence of BBD in DMD, the first sentence in paragraph 2 and the discussion that follows, emphasizes the importance in the selection of healthy controls for this is the crux of the communication. Therefore, the authors need to truly provide a detailed account of how they selected their control group in Methods.

Author Response

Response to Reviewer 2 Comments

Point 1: The abstract should be ‘structured’ with an Introduction, Methods, Results and Conclusions sections.

Response 1: We acknowledge this comment and structured the abstract as suggested. 

Point 2: Much of the 3rd paragraph in the Introduction of the main text (lines 72 – 86) can be shortened and transferred to the Methods section with the last part placed in the Discussion section.

Response 2: We acknowledge these suggestions and agree that these changes will improve the structure and herewith the quality of the manuscript. The first part of the third paragraph is moved to the methods section (2.2.2.1, p5) and the last part is placed in the discussion section (Clinical implications/recommendations, p18).

Point 3: In the Methods section under study design (2.2) please describe what chronological time period these questionnaires were distributed, exactly how the children with DMD and healthy controls were recruited initially, how many declined to participate, how many patients and families had to be excluded due to insufficient answering of the questions even though they were initially eligible?

Response 3: We agree with this comment, the methods could use better explanation. We added more detailed information on the study design and participants. Participants who met the inclusion criteria were asked by their treating physician or youth health care professional during regular visits if they would like to participate in this study and if the researcher could contact them about this. Only after saying yes, the questionnaires were sent to them by the researcher (JL), all sent questionnaires were returned. Unfortunately, we have no clear count of how many potential participants declined for each participating center as the deputies did not keep count. For my own center, CNL, only five patients, for whom regular appointments were planned during the 12 month-study duration met the inclusion criteria and all of these patients said yes to participation in this study. 

Information on how many participants and families had to be excluded due to insufficient answering even though they were initially eligible we added a statement to the results section (3.1 participant characteristics).

Point 4: How many patients answered the questionnaires themselves and how many caregivers had to respond to the questionnaires?

Response 4: All patients > 12 years answered the questionnaires themselves, for patients < 12 years old only parents/caregivers responded.

For DMD, n = 23 by proxy report, n = 34 self-report.

For HC, n = 30 by proxy report, n = 26 self-report.

I added these numbers to the results section. 

Point 5: In section 3.3.3 of the Results, I am not sure which group (DMD or HC or both) the first two sentences refer to. In sentence 2, “The other 31…” refers to which group? Can the authors clarify this point?

Response 5: We acknowledge this comment and clarified that we refer to the patient group in this section.

Point 6: Additionally, the number of patients quoted by the authors in this paragraph do not add up to the 57 DMD patients or the 56 healthy controls. Can the authors provide clarity to this point?

Response 6: We acknowledge this comment and therefore would like to clarify this. With ´all HC´ we mean the total HC group. To further clarify this we added the total number 56 of healthy controls. The reason that the numbers do not add up for the DMD group is because of a typo. Thank you so much for pointing this out. With ´the other´ I meant the remaining DMD patients minus the 16 patients, who independently use the toilet. I corrected 31 to 41. 

The 34 DMD patients using aids is, however, the correct number, because some patients did not use any aids but did need personal assistance and there are other patients that can use the toilet independently by using aids. In the additional questionnaire three questions about this topic were asked: do you use any aids, do you need an adjusted toilet, do you use the toilet independently? Which could lead to overlap in numbers. This questionnaire was developed and used in this study for the first time, and should be optimized for future use after putting it in practice. 

Point 7: In section 3.3.4 of the Results, were there any age adjustment differences noted between the two groups and if so, please describe them in a revised version.

Response 7: Since we found a higher median age in the total DMD group compared with HC (p = .006), it is not possible to differentiate if a higher median age of a negative impact of BBD on daily life is a result of the overall higher age of the cohort or if a negative impact of BBD on daily life is more common at a higher age for DMD patients. This is discussed in the discussion section (prevalence of bbd in dmd). For the revised version, we also added a statement on this topic to the Impact of BBD on quality of life paragraph of the Discussion section.

Point 8: In the Discussion section, under prevalence of BBD in DMD, the first sentence in paragraph 2 and the discussion that follows, emphasizes the importance in the selection of healthy controls for this is the crux of the communication. Therefore, the authors need to truly provide a detailed account of how they selected their control group in Methods.

Response 8: We acknowledge this comment and added a more detailed description on the selection of the control group in the Methods section.